# Accuracy of self-diagnosis in conditions commonly managed in primary care: diagnostic accuracy systematic review and meta-analysis

Julie McLellan ●,[1] Carl Heneghan ●,[1] Nia Roberts,[2] Annette Pluddemann ●[1]

¹Nuffield Department of Primary Care Health Sciences, University of Oxford, Oxford, UK
²Bodleian Health Care Libraries, University of Oxford, Oxford, UK

**Correspondence to**
Dr Annette Pluddemann;
annette.pluddemann@phc.ox.ac.uk

## ABSTRACT

**Objectives** To assess the diagnostic accuracy of self-diagnosis compared with a clinical diagnosis for common conditions in primary care.

**Design** Systematic review. Meta-analysis.

**Data sources** Medline, Embase, Cochrane CENTRAL, Cochrane Database of Systematic Reviews and CINAHL from inception to 25 January 2021.

**Study selection** Eligible studies were prospective or retrospective studies comparing the results of self-diagnosis of common conditions in primary care to a relevant clinical diagnosis or laboratory reference standard test performed by a healthcare service provider. Studies that considered self-testing only were excluded.

**Data extraction** Two authors independently extracted data using a predefined data extraction form and assessed risk of bias using Quality Assessment of Diagnostic Accuracy Studies-2.

**Methods and results** 5047 records identified 18 studies for inclusion covering the self-diagnosis of three common conditions: vaginal infection (five studies), common skin conditions (four studies) and HIV (nine studies). No studies were found for any other condition. For self-diagnosis of vaginal infection and common skin conditions, meta-analysis was not appropriate and data were reported narratively. Nine studies, using point-of-care oral fluid tests, reported on the accuracy of self-diagnosis of HIV and data were pooled using bivariate meta-analysis methods. For these nine studies, the pooled sensitivity was 92.8% (95% CI, 86% to 96.5%) and specificity was 99.8% (95% CI, 99.1% to 99.9%). Post hoc, the robustness of the pooled findings was tested in a sensitivity analysis only including four studies using laboratory testing as the reference standard. The pooled sensitivity reduced to 87.7% (95% CI, 81.4% to 92.2%) and the specificity remained the same. The quality of all 18 included studies was assessed as mixed and overall study methodology was not always well described.

**Conclusions and implications of key findings** Overall, there was a paucity of evidence. The current evidence does not support routine self-diagnosis for vaginal infections, common skin conditions and HIV in primary care.

**PROSPERO registration number** CRD42018110288.

## STRENGTHS AND LIMITATIONS OF THIS STUDY

⇒ This systematic review summarises and interprets the available evidence on self-diagnosis of conditions managed in primary care.

⇒ This search strategy was extensive including publications identified from databases Medline, Embase, Cochrane Central Register of Controlled Trials (CENTRAL), Cochrane Database of Systematic Reviews and CINAHL, up to January 2021.

⇒ Standard methodology for systematic review of diagnostic accuracy studies was used, including study quality appraisal using Quality Assessment of Diagnostic Accuracy Studies-2.

⇒ There was a paucity of evidence for many common conditions.

⇒ Lack of evidence meant meta-analysis was possible only for one condition.

## INTRODUCTION

The workload in primary care continues to increase,[1 2] in part not only due to global population increases but also due to age of populations, change in lifestyle and more complex health problems.[3] Increased workload has been recognised in the UK as an important factor in working conditions for primary care physicians (general practitioners, GPs) and strategies are required to manage the workload.[4]

One strategy is the potential for self-diagnosis and self-treatment by patients of some commonly occurring conditions. If feasible, this could lead to more rapid diagnosis and treatment reducing the burden on primary care services. The prospect of self-diagnosis is controversial with concerns if results are misinterpreted or patients fail to confirm their findings to a physician.[5] In terms of the evidence, the first important question is to assess the diagnostic accuracy of self-diagnosis in the primary care setting. Subsequently, in order to support self-diagnosis, the efficacy needs to be assessed, to

inform which conditions can be self-diagnosed safely, in which circumstances and by whom.

Cooke *et al* recently reported the 30 most commonly managed conditions in primary care in Australia, which has a health landscape broadly comparable with western Europe.[6] This list arises from survey data collected between January 2009 and December 2010, which included 194100 patient encounters from 1941 GPs.

This systematic review, therefore, aimed to assess the diagnostic accuracy of self-diagnosis compared with a clinical diagnosis for common conditions in primary care by a healthcare provider.

## METHODS

### Types of studies
We included prospective or retrospective studies comparing the results of self-diagnosis of common conditions in primary care to the results of a relevant clinical diagnosis or laboratory reference standard test. We excluded studies with a case–control design due to their high risk of bias.[4]

### Population
The included population was adults self-diagnosing common conditions in primary care. Common conditions included were broadly based on those reported by Cooke *et al*[6] and relevant for self-diagnosis (see online supplemental table 1). Studies in children, based in animals or non-human samples were excluded.

### Index test
The index test was self-diagnosis, where we defined 'self-diagnosis' as a diagnosis made by the patients in the study, including self-evaluation and interpretation of results of rapid tests. Studies that considered self-testing only and not as part of self-diagnosis were excluded, in addition studies assessing accuracy of self-monitoring of an existing condition were excluded.

### Reference standard
The reference standard was clinical diagnosis or laboratory test performed by a healthcare service provider. We excluded studies comparing self-diagnosis with diagnosis by allied health professionals or pharmacists.

### Outcome measures
To be included in the review, studies must have reported diagnostic accuracy measures (eg, sensitivity, specificity, likelihood ratios, predictive values, etc) and primary data for 2×2 tables. We excluded studies reporting only measures of agreement.

### Search methods to identify studies
The search strategy was based on a combination of terms for self-testing and self-diagnosis, diagnostic accuracy terms (e.g., sensitivity, specificity, etc.) and terms for common conditions in primary care[6] (see online supplemental table 2 for full search strategy).

We searched the following electronic databases from inception to 25 January 2021: Medline (OvidSP) (1946–present), EMBASE (OvidSP) (1974–present), Cochrane Central Register of Controlled Trials (CENTRAL) and Cochrane Database of Systematic Reviews via Cochrane Library, Wiley) (Issue 1 of 12 January 2021) and CINAHL (EBSCOHost) (1982–present). No restrictions were imposed on study population numbers or language (studies in languages other than English were translated). Letters, narrative reviews and other non-primary sources were excluded. The reference lists of included studies, plus the first five 'similar articles' identified through PubMed for these studies, and reference lists of relevant systematic reviews were used to identify further relevant publications. References were imported into Endnote X9[7] where duplicates were removed.

### Data collection and analysis

#### Selection of studies
Two reviewers independently applied the selection criteria to the titles and abstracts of the study reports identified by the searches. Full text of all studies that met the inclusion criteria were reviewed to agree the final list of included studies. Disagreements between reviewers were resolved by discussion and where agreement could not be reached a third reviewer was consulted.

#### Data extraction and management
Two reviewers (JM, AP) independently extracted information from selected studies into a predefined data extraction sheet (see online supplemental table 3) and crosschecked the data. Disagreements were resolved by discussion.

### Assessment of methodological quality
We used the Quality Assessment of Diagnostic Accuracy Studies-2 (QUADAS-2)[8] tool to assess methodological quality of included studies. This considered the risk of bias in four domains (patient selection, index test, reference standard, flow and timing), as well as assessing the applicability (for the first three domains) of the studies to the review research question. Studies were assessed as low, high or unclear risk of bias/concerns regarding applicability for each domain. Two reviewers (JM, AP) independently assessed studies' methodological quality; disagreements were resolved by discussion, or if necessary, by a third reviewer. The results of the QUADAS-2 assessment were presented in a summary table.

### Statistical analysis and data synthesis
Data were presented and analysed based on the condition being diagnosed. We compiled summary tables outlining the detailed study information of included studies, including the patient sample, condition, study design, setting, the test under evaluation, the comparator and conduct of the study. We extracted binary diagnostic accuracy data from all studies and constructed 2×2 tables.

## Meta-analysis

We used Review Manager[9] to produce paired forest plots to explore the between-study variability of sensitivity and specificity across the included studies. For each study estimate of sensitivity and specificity, corresponding 95% CIs were shown to illustrate the uncertainty related to each study estimate. Where different thresholds were applied these were reported. Where appropriate, we used bivariate meta-analysis methods[10] to generate pooled estimates of sensitivity and specificity. Due to the nature of the data, a change was made to the protocol and RStudio[11] was used to generate the model parameters to input into Revman.[9]

## Investigating heterogeneity

For medical conditions for which data from more than one study was available and where it was possible to investigate between-study heterogeneity in the results, inclusion of study level characteristics as covariates in meta-analysis and subgroup analyses were considered. These approaches were carried out if there was sufficient data available and subgroup specific pooled estimates were thought to be of clinical relevance.

## Investigating reporting bias

Funnel plots used to detect publication bias in reviews of randomised controlled trials (RCTs) have been shown to be misleading for diagnostic test accuracy reviews.[12 13] Funnel plots as an assessment of reporting bias were therefore not be included in this review.

## Patient involvement

Members of the public were part of the research programme committee of the National Institute for Health Research (NIHR) programme grant that funded this study. Updates and details about the study were presented to the committee while the study was ongoing, and the public members provided feedback. This review formed part of the NIHR Evidence Synthesis Working Group (ESWG) and members of the public who were part of the ESWG steering committee commented on the protocol for the study and on updates presented to the steering committee.

The full protocol is provided in online supplemental table 4.

## RESULTS

Figure 1 shows a summary of the search results and the inclusion and exclusion of studies. After removal of duplicates, 5047 records were identified through database searches, websites and citation searching. This resulted in full texts of 170 articles being assessed for eligibility and

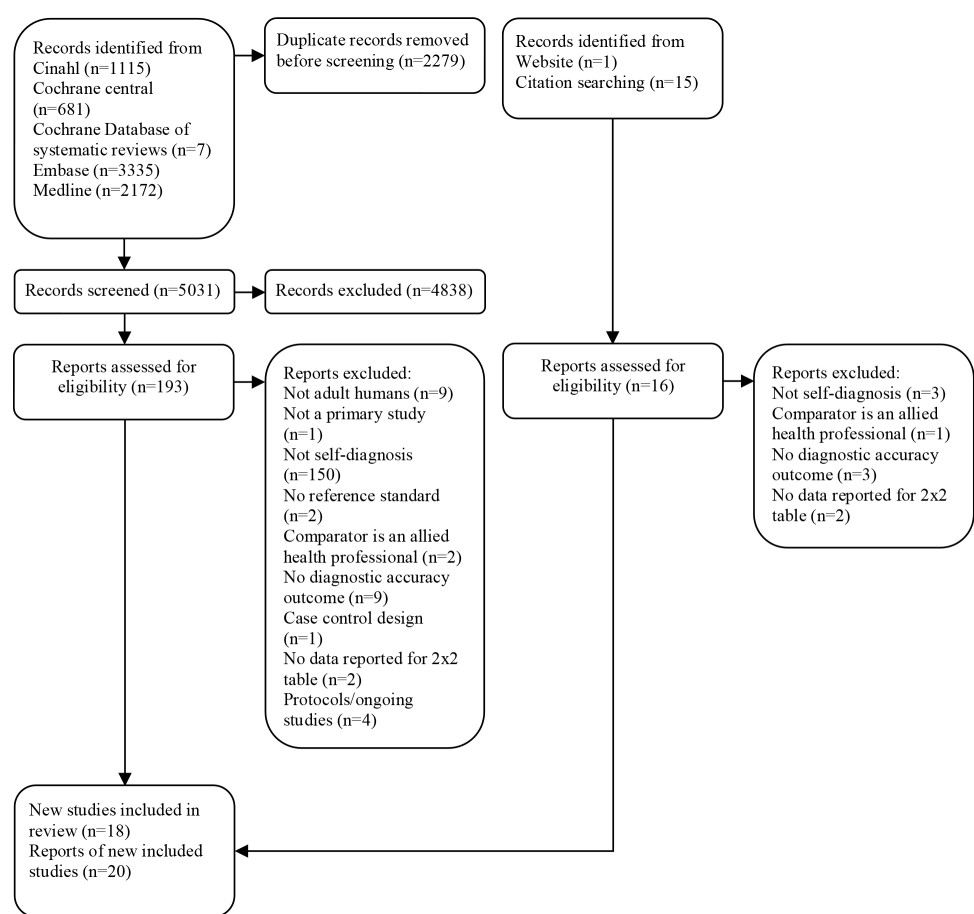

**Figure 1**   Study selection.

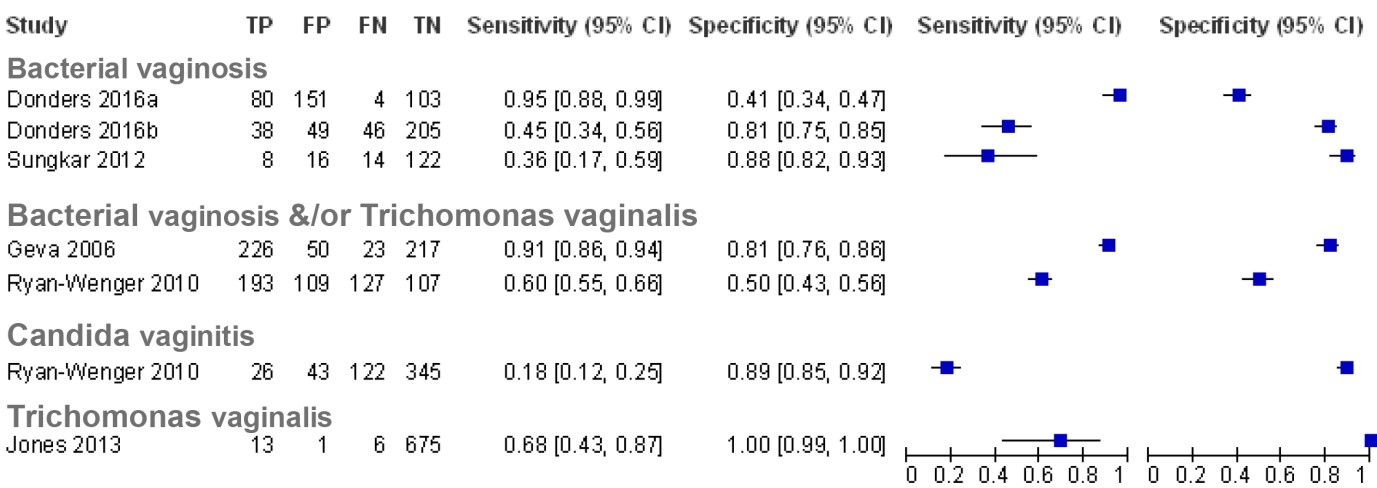

**Figure 2** Paired forest plots of sensitivity and specificity for studies of self-diagnosis of vaginal infection (where Donders[14] 2016a used pH threshold≥4.5 and Donders[14] 2016b used pH threshold≥4.7. Sungkar *et al* represent pooled data from four time points[15]). TP, true positives; FP, false positives; FN, false negatives; TN, true negatives.

20 included articles[14–33] reporting results of 18 individual studies.[14–20 22–28 30–33] These 18 included studies fell into three broad groups of commonly managed conditions as defined by Cooke et al: "Female genital infection", "dermatitis and contact/allergic", and "viral disease and not otherwise specified". No studies of self-diagnosis were found for any other conditions in primary care.

Most excluded studies only reported on patients' ability to self-test (or self-monitor an existing condition) with the diagnosis being made by a clinician and did not report diagnostic accuracy of self-diagnosis.

Of the included studies, five reported on the accuracy of self-diagnosis of vaginal infection,[11–15] four for common skin conditions[19 20 22 23] and nine for the self-diagnosis for HIV.[24–28 30–33] Online supplemental tables 5 and 6 summarise the characteristics of included studies and characteristics of self-diagnostic (index) and reference tests, respectively. Paired plots of sensitivity and specificity were generated, grouping the studies by the condition to be diagnosed (figures 2–4). For studies examining the accuracy of self-diagnosis of vaginal infection and in common skin conditions,[14–20 22 23] meta-analysis was not appropriate due to the between-study heterogeneity and

the overall low number of studies, which would make meta-analysis uninformative.[34]

### Self-diagnosis of vaginal infections

Five studies assessed the accuracy of self-diagnosis of bacterial vaginosis and/or infection with *Trichomonas vaginalis*,[14–18] with one study assessing the self-diagnosis of *Candida* vaginitis[17] (figure 2). For bacterial vaginosis, the accuracy of a vaginal fluid test using a pH strip was assessed with laboratory testing (Gram staining) as the reference standard in two studies[14 15] (online supplemental table 5). For the diagnosis of bacterial vaginosis and/or *T. vaginalis*, a panty liner test kit (VI-SENSE) for vaginal discharge was assessed against a combination of clinical and laboratory assessment as the reference test in one study,[16] and a vaginal fluid self-diagnosis kit for women in the military was assessed with clinical and laboratory assessment as the reference test in the second study.[17] One study used a vaginal fluid dipstick test for the presence of *T. vaginalis* (OSOM Trichomonas rapid test) compared with a laboratory PCR as a reference test.[18] For the self-diagnosis of *Candida* vaginosis, a military self-testing kit based on a combination of the measurement of

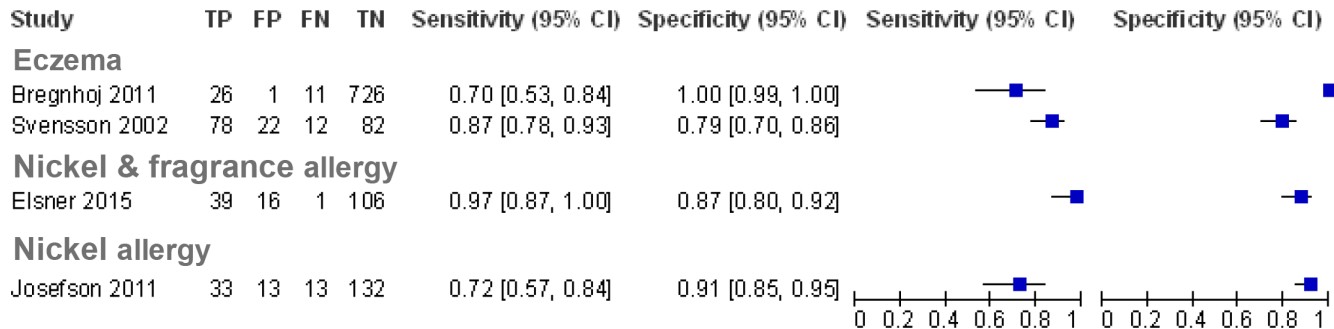

**Figure 3** Paired forest plots of sensitivity and specificity for studies of self-diagnosis of common skin conditions (where Bregnhoj *et al*[19] represent pooled data from recruitment and follow-up time points). TP, true positives; FP, false positives; FN, false negatives; TN, true negatives.

**A**

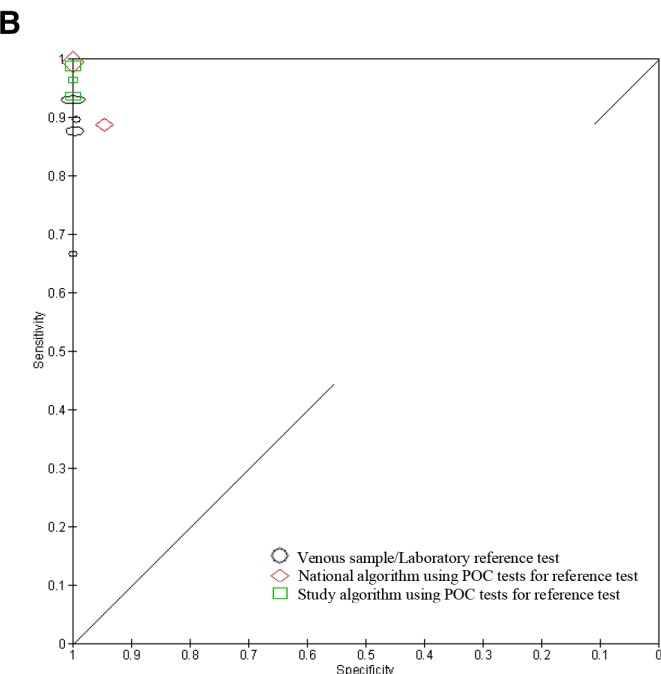

| Study | TP | FP | FN | TN | Sensitivity (95% CI) | Specificity (95% CI) |
|---|---|---|---|---|---|---|
| Assiimwe 2014 | 24 | 5 | 3 | 91 | 0.89 [0.71, 0.98] | 0.95 [0.88, 0.98] |
| Belete 2019 | 199 | 0 | 1 | 200 | 0.99 [0.97, 1.00] | 1.00 [0.98, 1.00] |
| Choko 2011 | 27 | 0 | 1 | 210 | 0.96 [0.82, 1.00] | 1.00 [0.98, 1.00] |
| Choko 2015 | 132 | 1 | 9 | 1507 | 0.94 [0.88, 0.97] | 1.00 [1.00, 1.00] |
| Kapaku 2017 | 227 | 7 | 32 | 2286 | 0.88 [0.83, 0.91] | 1.00 [0.99, 1.00] |
| Kurth 2016 | 26 | 1 | 3 | 173 | 0.90 [0.73, 0.98] | 0.99 [0.97, 1.00] |
| Martinez Perez 2016 | 323 | 0 | 4 | 1860 | 0.99 [0.97, 1.00] | 1.00 [1.00, 1.00] |
| Orasure 2012 | 106 | 1 | 8 | 5385 | 0.93 [0.87, 0.97] | 1.00 [1.00, 1.00] |
| Pant Pai 2013 | 6 | 0 | 3 | 242 | 0.67 [0.30, 0.93] | 1.00 [0.98, 1.00] |

**B**

**Figure 4** Studies of self-diagnosis of HIV. (a) Paired forest plots of sensitivity and specificity (where Choko et al[27] represent pooled data from 1–12 to 13–24 months follow-up time points). (b) Receiver operating characteristic plot of HIV self-diagnosis compared with clinical diagnosis or laboratory reference test grouped by reference test type (where size of symbol indicates study size).

pH, amines and the symptom of vaginal itching for self-diagnosis was compared with a combination of clinical and microbiological laboratory assessment in one study.[17]

### Bacterial vaginosis and/or *T. vaginalis*
The sensitivity of a self-taken swab applied to a pH test strip for self-diagnosis of bacterial vaginosis ranged from 0.45 (95% CI, 0.34 to 0.56)[14] to 0.60 (95% CI, 0.55 to 0.66)[17] at a pH cut-off of ≥4.7. Notably, Ryan-Wenger et al,[17] the study reporting a sensitivity of 0.60, also included symptoms (vaginal itching) and the presence of amines as part of the assessment, which may explain the higher sensitivity. Donders et al[14] assessed a lower cut-off of pH≥4.5 and showed an increased sensitivity of 0.95 (95% CI, 0.88 to 0.99), however at the expense of specificity. The study on pregnant women[15] assessing the accuracy of pH test strips did not specify the pH cut-off and reported a sensitivity of 0.36 (95% CI, 0.17 to 0.59) with a specificity of 0.88 (95% CI, 0.82 to 0.93). The specificity for

the pH test strip tests ranged from 0.5 (95% CI, 0.43 to 0.56)[17] to 0.81 (95% CI, 0.75 to 0.85)[14] at the pH cut-off of ≥ 4.7, decreasing to 0.41 (95% CI, 0.34 to 0.47) at the lower cut-off of pH≥4.5.[14] Interestingly, the low specificity of 0.5 was reported by the study combining the pH test strip with symptoms and the presence of amines.[17]

The study assessing the vaginal discharge test using a panty liner test kit with an indicator strip incorporated in the liner[16] reported a sensitivity of 0.91 (95% CI, 0.86 to 0.94) and specificity of 0.81 (95% CI, 0.76 to 0.86) for the diagnosis of bacterial vaginosis and/or *T. vaginalis* infection.

### *T. vaginalis*
One study in Brazil assessed a rapid immunochromatographic *T. vaginalis* test for use at home[18] and reported a sensitivity of 0.68 (95% CI, 0.43 to 0.87) and specificity of 1.00 (95% CI, 0.99 to 1.00) for self-diagnosis of *T. vaginalis* infection.

### *Candida* vaginitis

Only one study[17] specifically assessed the diagnostic accuracy of self-diagnosis of *Candida* vaginitis, which formed part of the military self-testing kit, and reported a sensitivity of 0.18 (95% CI, 0.12 to 0.25) and specificity of 0.89 (95% CI, 0.85 to 0.92).

### Self-diagnosis of common skin conditions

Four studies assessed the accuracy of self-diagnosis of common skin conditions[19 20 22 23] (figure 3). We included two studies that were outside our age inclusion criteria: Bregnhoj[19] 2011 reported patients included had a mean age 17.5 years, nevertheless these patients would have been 16+ years to qualify as apprentice hairdressers. And Svensson *et al*[20] reported the mean age of patients as 40.4 years (no SD), but included patients from age 16 years.

### Eczema

Two studies assessed the accuracy of self-diagnosis based on a self-evaluated questionnaire of signs and symptoms for the diagnosis of eczema alongside a self-assessment of the presence or absence of eczema based on the questionnaire results, compared with assessment by a clinician.[19 20] Overall, 710 participants were included across the two studies. The reported sensitivity ranged from 0.7 (95% CI, 0.53 to 0.84) to 0.87 (95% CI, 0.78 to 0.93) and specificity ranged from 0.79 (95% CI, 0.70 to 0.86) to 1.00 (95% CI, 0.99 to 1.00). The relatively high specificity suggests the potential for patients to use the questionnaire as a tool to confirm that they have eczema and therefore seek healthcare advice; however, it should be noted that there was an unclear risk of bias regarding the patient selection. In Svensson *et al*,[20] the patients were recruited at a dermatology outpatient clinic, where they had been referred to and 113 patients in the study reported having had a diagnosis of eczema in the last 12 months, suggesting a more selected population with a higher pretest probability. While the setting in the study by Bregnhoj *et al*[19] was not reported, it was conducted among hairdressers who may have had more experience of eczema either themselves or colleagues being diagnosed with the condition, given the nature of their profession. They may be more aware of the signs and symptoms and may constitute a selected population. Therefore, the diagnostic accuracy of the questionnaire for self-diagnosis may be dependent on the type of population.

### Skin allergy

Two studies assessed the diagnostic accuracy of self-diagnosis to detect an allergic skin reaction including 408 participants across the two studies; one study assessed nickel and/or fragrance allergy,[22] and the other assessed nickel allergy alone.[23] Both studies used a patch test applied to the arm, which was self-evaluated by participants 2–4 days later. Dermatologists then also evaluated the patch tests as the reference standard. One study recruited participants at hospital dermatology departments,[23] while the other recruited through a newspaper advertisement targeted at people with a self-suspected allergy towards fragrance and/or nickel.[22] Sensitivity ranged from 0.72 (95% CI, 0.57 to 0.84) to 0.97 (95% CI. 0.87 to 1.00) and specificity ranged from 0.87 (95% CI, 0.80 to 0.92) to 0.91 (95% CI, 0.85 to 0.95). Elsner *et al*[22] also reported that participants found the information regarding how to apply the test extensive and detailed; the information regarding self-evaluation of the test was limited and should be improved.

### Self-diagnosis of HIV

Nine studies were identified that reported the diagnostic accuracy of self-testing and self-diagnosis of HIV.[24–28 30–33] In all studies, self-diagnosis was undertaken unsupervised using a rapid point-of-care (POC) oral fluid test manufactured by OraSure Technologies, either OraQuick In-Home intended for lay users or OraQuick Advance intended for professional use. The studies recruited 13 103 participants, and all studies were conducted in African countries except for the phase III trial in the USA by OraSure Technologies.[33] The 2019 global HIV prevalence rates for women and men aged 15–49 were 0.8% (95% CI, 0.7 to 1.0) and 0.6% (95% CI, 0.5 to 0.8), respectively, with the overall highest prevalence by country in Eswatini (Africa) at 27.1% (95% CI, 25.4 to 28.8).[35] All included studies had prevalence rates above the global averages for men and women or, if not reported, were in countries with high prevalence rates. Prevalence rates ranged from 2.12% in the USA study[33] to 22.1% in the Ugandan study.[24] The USA study was conducted in 20 clinical sites, 17 identified as high prevalence sites (2.6%) and 3 as low prevalence sites (0.1%). All studies enrolled participants from the general population including the USA where no breakdown of sexual orientation was reported. The reported sensitivity and specificity were similar between studies (figure 4a); the single study[32] reporting a lower estimate for sensitivity still had a CI that overlapped with half of the other studies. The pooled sensitivity based on all 9 included studies was 92.8% (95% CI, 86% to 96.5%) and the pooled specificity was 99.8% (95% CI, 99.1% to 99.9%). The studies showed low heterogeneity (figure 4b). The reference standard used in the studies was one of the three types: four studies took a venous sample which was sent to a laboratory for testing,[28 30 32 33] two studies used a nationally approved algorithm based on a combination of rapid POC tests[24 25] and three studies used a study based algorithm again based on rapid POC tests.[26 27 31] In three studies using POC tests for the reference standard, the diagnosis may have been by clinicians, but it was unclear. These studies reported the diagnosis by a research assistant[24] or a counsellor.[26 31] Post hoc, a sensitivity analysis was conducted to test the robustness of the pooled findings by removing the studies using POC tests as the reference standard (including tests where it was unclear whether diagnosis was by a clinician) and only including those studies using laboratory testing. Based on four studies using laboratory testing as the reference standard,[28 30 32 33] the pooled sensitivity was 87.7%

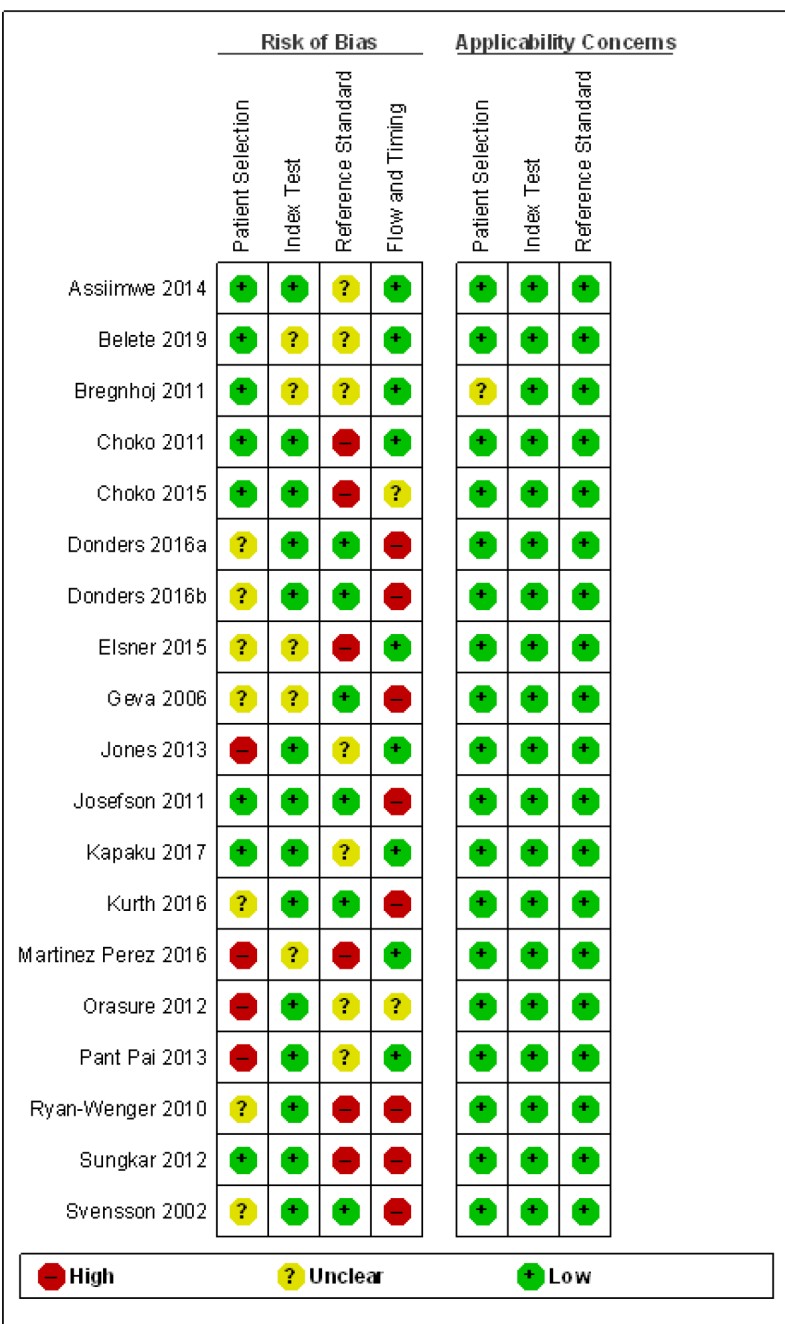

**Figure 5** Quality Assessment of Diagnostic Accuracy Studies-2 summary of risk of bias and applicability concerns showing review authors' judgements about each domain for each included study (based on 18 studies (19 data sets)).

(95% CI, 81.4% to 92.2%) and the pooled specificity was 99.8% (95% CI, 98.9% to 99.9%). No data were reported by participant characteristics such as gender, ethnicity or sexual orientation.

A number of studies[24–26 30–32] reported on the viability or feasibility of the oral fluid self-test stating that participants found it easy to conduct, but acknowledged that instructions should be adapted to the population using the test, particularly the literacy levels. Furthermore, users must be encouraged to receive a confirmatory test.

### Methodological quality of included studies

Assessment of the quality of included studies using the QUADAS-2 framework[8] is presented in figure 5, which summarises the overall risk of bias and applicability concerns. For patient selection, the risk of bias overall was mixed; where studies were rated unclear in this domain, it was because several studies either recruited a selected population with a potentially higher pretest probability or they did not clearly report the recruitment strategy and whether eligible patients were consecutively recruited. However, with the exception of one study, applicability concerns were low. In several cases, although the population may have been a selected population (eg, at high risk of HIV infection or skin eczema or with a prior history of eczema or vaginal infection), it could be argued that these might be the populations where self-diagnosis and/

or self-testing may be most relevant. Overall risk of bias regarding the conduct of the index test was low and in most studies participants were blinded to the results of the reference test, though in some studies this was not clearly described. Risk of bias with respect to the reference test was unclear or high in some studies as assessors were either not blinded to the results of the index test or blinding was unclear. For the domain of flow and timing, several studies were judged to be at high risk of bias as the interval between testing was frequently not explicitly reported. In addition, differential reference bias was identified as present in some studies and unclear in others, in particular it was unclear in several studies how clinical assessment was conducted or standardised. Overall study procedures were not always clearly described.

## DISCUSSION

With the increasing workload in primary care[1] and the continued development of rapid tests, including those that are intended to be used by patients, we aimed to assess the evidence for the diagnostic accuracy of self-diagnosis. We identified limited evidence on the diagnostic accuracy of self-diagnosis: only 20 publications (reporting data from 18 studies) specifically assessed the accuracy of self-diagnosis, covering three commonly managed conditions, namely, vaginal infections, common skin conditions and HIV. Interestingly, no studies of self-diagnosis were found for any other conditions in primary care. It was particularly notable that we did not find any studies assessing the diagnostic accuracy of self-diagnosis of common primary conditions such as upper respiratory tract and urinary tract infections. As technology develops potentially enabling increased self-diagnosis in primary care, we would expect future reviews examining this research question to include more common conditions. In particular, we would expect to include studies for self-diagnosis of COVID-19 following the rapid development of tests during the COVID-19 pandemic.

The evidence for self-diagnosis of vaginal infection suggests a lack of sufficient accuracy to aid self-diagnosis. Tests relying on self-swabs and pH strips showed low sensitivity (below 60%) and therefore would not be useful to rule out disease. Although sensitivity improved to 95% with increasing pH cut-off, this occurred at the cost of specificity, which dropped from 81% to 41%. Using the test at this cut-off would, therefore, result in concern for missed diagnoses. The immunochromatography test also showed insufficient sensitivity (68%) to be of use as a rule-out test. The panty liner test, however, had high sensitivity of 91% and may prove to be a useful rule-out test, although this result is limited as it is based on one study judged to be at unclear risk of bias, with a high risk of bias in the flow and timing domain. In terms of their use to confirm the presence of vaginal infection, the highest specificity (100%) was reported for the immunochromatographic test. This may be a useful test to aid self-diagnosis of vaginal infection in systems that currently rely on syndromic management, such as low resource settings, particularly to improve the targeting of antimicrobial prescribing. However, it should be noted this result is also based on one study and requires confirming in a larger study.

The allergy patch tests overall showed reasonable accuracy (sensitivity 72%–97%; specificity 87%–91%) and may be useful as an initial self-screening test for patients with a suspected nickel or fragrance allergy; however, it could be argued that the main use of these tests might be to safely rule out a contact allergy. The reported diagnostic accuracy suggests the tests are not sufficiently sensitive as a rule-out test, particularly given the relatively wide CIs. The tests may however be useful in settings where access to dermatology services is scarce. The self-diagnosis of eczema using a questionnaire of signs and symptoms showed specificity ranging from 0.79 to 1.00, suggesting this test might be useful as a confirmation test for patients for suspect they have eczema and to then seek treatment and management advice. The selected nature of the patients in the studies (ie, patients with a previous eczema diagnosis and hairdressers, who may encounter eczema more frequently due to their profession) may overestimate the accuracy of the test; however, it could be argued that these might be the populations in whom the test is most relevant.

We identified nine studies that reported the accuracy of self-diagnosis of HIV (sensitivity 93%, specificity 99%). However, the sensitivity is reduced to 88% when only studies using a venous sample and laboratory testing as the reference standard are included. With a sensitivity of 88%, the accuracy data would not support the use of this test as a rule-out test, particularly given the clinical consequences of a false negative test result. Evidence suggests there may be benefits to self-initiated HIV testing, including early identification, increased likelihood for the uptake of HIV prevention interventions and a reduction in sexual risk behaviours,[36] warranting further research, in particular in resource-limited settings where access to testing sites may be a barrier. However, HIV self-testing should also be considered in the context of linkage to care, access to counselling and adequate regulatory and quality assurance systems.[37]

The search strategy for this review was broad and extensive with few restrictions resulting in the high number of publications to screen. While it is possible, it is unlikely, studies were missed. We are unaware of any other reviews examining this research question. The main limitation was the lack of available evidence for a number of common conditions; studies reporting on self-testing alone were more common, but few studies assessed self-diagnosis, with patients interpreting the test results and making a diagnosis independently. For the three conditions where we identified studies reporting on the diagnostic accuracy of self-diagnosis, there was a paucity of evidence. Many studies were not replicated and included small sample sizes and contained methodological biases that limited the application of the results to

practice. For self-diagnosis of vaginal infections, common skin conditions and HIV, further research is required to draw a definitive conclusion on the efficacy of self-diagnosis. For other common conditions in primary care, research is needed on self-diagnosis where this option is available, and studies should go beyond considering self-testing alone and also assess the diagnostic accuracy of self-diagnosis. Terminology for self-diagnosis, self-testing and self-screening is overlapping in some cases and needs clarifying. Finally, research is required into the patient's readiness and attitude towards self-diagnosis along with its effect on the patient/physician relationship.

The current limited evidence does not support routine self-diagnosis for vaginal infections, common skin conditions and HIV in primary care.

**Acknowledgements** The authors would like to thank Hayley Jones for joint authorship of the original protocol, Elizabeth Spencer for assistance with preparation of the protocol and Emily McFadden for assistance with screening of studies for inclusion.

**Contributors** AP and CH conceived the idea for this review. AP led the research, developed the protocol and conducted the review (screening and extraction). CH developed the protocol and provided clinical input. NR devised and conducted the search strategy and approved the manuscript. JM conducted the review (screening, extraction, data analysis). JM, AP and CH jointly prepared the manuscript. AP is responsible for the overall content as the guarantor.

**Funding** This project was funded by National Institute for Health Research School for Primary Care Research (Project Number 390).

**Competing interests** JM reports grants from National Institute for Health Research School of Primary Care Research (NIHR SPCR) (Evidence Synthesis Working Group Project Number 390) and part funding from the Thames Valley Applied Research Collaborative, during the conduct of the study, and occasionally receives expenses for teaching evidence-based medicine. AP reports grants from NIHR SPCR (Evidence Synthesis Working Group Project Number 390), during the conduct of the study, and occasionally receives expenses for teaching evidence-based medicine. CH reports receiving expenses and fees for his media work. He has received expenses from the WHO and holds grant funding from the NIHR, the NIHR SPCR, the Wellcome Trust and the WHO. He has received financial remuneration from an asbestos case. He has also received income from the publication of a series of toolkit books published by Blackwell. On occasion, he receives expenses for teaching evidence-based medicine (EBM) and is also paid for his general practitioner work in NHS out of hours. The Centre for Evidence-Based Medicine (CEBM) jointly runs the EvidenceLive Conference with The British Medical Journal (BMJ) and the Overdiagnosis Conference with some international partners which are based on a non-profit making model.

**Patient and public involvement** Patients and/or the public were involved in the design, or conduct, or reporting, or dissemination plans of this research. Refer to the Methods section for further details.

**Patient consent for publication** Not applicable.

**Ethics approval** Not applicable.

**Provenance and peer review** Not commissioned; externally peer reviewed.

**Data availability statement** All data relevant to the study are included in the article or uploaded as supplementary information. All data are available in published articles.

**ORCID iDs**
Julie McLellan http://orcid.org/0000-0002-2868-8631
Carl Heneghan http://orcid.org/0000-0002-1009-1992
Annette Pluddemann http://orcid.org/0000-0003-2101-0390

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
