## [Reviewer comments · BMJ Open]

ARTICLE DETAILS

TITLE (PROVISIONAL)	Accuracy of self-diagnosis in conditions commonly managed in primary care: Diagnostic accuracy systematic review and meta-analysis
AUTHORS	McLellan, Julie; Heneghan, Carl; Roberts, Nia Pluddemann, Annette

VERSION 1 – REVIEW

REVIEWER	Kerdelmidis, Melissa Canterbury District Health Board, Planning & Funding
REVIEW RETURNED	20-Dec-2021

GENERAL COMMENTS	A good study. To make it even more robust, suggest add a couple of sentences: 1. In the Results, HIV self tests: pls clarify the population group the US HIV study was in, as the pre test probability of HIV varies among different population groups eg Men who have Sex with Men (MSM) compared to general American population.2. Equity in testing populations - A sentence or two in the Results would be useful, commenting on whether the self tests worked the same for everyone or better for some groups. Were there any differences by gender or ethnicity that were noted or striking among the tests? If this information wasn't in the studies, it may be worth mentioning that it was missing. Thank you.
---

REVIEWER	Yonemoto, Naohiro Kyoto University School of Public Health, Biostatistics
REVIEW RETURNED	21-Dec-2021

GENERAL COMMENTS	This is a systematic review and meta-analysis for accuracy of self-diagnosis in conditions commonly managed in primary care. I have some comments. 1. I am not clear why you set the object in the study. Really Do self-diagnosis an self-treatment (this is out of scope in the study) is useful strategy for any disease ? You should more clearly describe the rationale of the objective with citations.2. The review have so comprehensive range, but the findings were limited. You should more clearly limit it by PICO.3. GP system as UK and Australia is not in global system. if the target is on the GP, you should limit the range of country.
---

	4. L24, The sentence is not clear. Is this a part of methods ? delete ? 5. L33, How to clarify the study design as "prospective or retrospective" ? 6. The setting in eligible is not clear. GP only or not ? 7. The eligible studies have many type of disease, because the search did not clearly set the objects by PICO. You should more clearly set the objects as a review.
--	---

REVIEWER	Campbell, John University of Exeter, Primary Care
REVIEW RETURNED	07-Mar-2022

GENERAL COMMENTS	Accuracy of self-diagnosis of condition commonly managed in primary care Thank you for the opportunity to review this systematic review and meta-analysis provided by colleagues from Oxford. The broad area of interest is of importance and the findings will be of interest to a wide general readership, especially readers from primary care.  • The abstract is structured, concise and well presented. Risk of bias has been assessed in a structured way. Whilst the study sets out to identify “common conditions” in primary care, I was somewhat surprised at the emphasis on self-diagnosis of HIV. There appeared to be no focus on some of the most common conditions encountered in primary care - upper-respiratory infection, urinary tract infection, impetigo etc. The use of comparison with lab reference data may account for some of this, but I had difficulty in respect of “clinical diagnosis” in seeing that the three conditions outlined are indeed worthy of the description attributed. Sophisticated statistics have been applied but the conclusion drawn is surprising, suggesting as it does that ‘routine self-diagnosis for common conditions in primary care is not appropriate’. • The introduction is very short, relating the issue of self-diagnosis to working conditions for general practitioners. The authors base their work on a primary care study from Australia – it is not clear why a wider range of literature has not been explored, for example relating to CPRD or the national survey of primary care morbidity which although dated would provide potential important useful information. The authors cite reference 3 as key to their background – but surprisingly, the ten most commonly identified conditions in that study do not form the bases of this work. I found this surprising. • Methods – A range of inclusion/exclusion criteria relating to the type of study and focus on clinical diagnosis or laboratory reference standard were included. An appropriate search methodology has been outlined, and would allow for reproducibility of the method. A clear outline of the approach to data collection and analysis, assessment of methodological quality and relevant statistical analysis/data synthesis is provided.  • The results are clearly presented, and well summarised in figure 1. From 5047 records, 18 individual studies were ultimately summarised. Whilst the authors provide the search strategies adopted in each of the databases searched, it is not clear to me exactly how the resulting conditions came to be focussed on.
--

	 • For the conditions investigated, a comprehensive and standardised approach has been adopted to reporting the results of self-diagnosis with relevant laboratory findings. In fact, it looks to me as though in respect of vaginal infections it is self-testing that has been investigated not self-diagnosis which is a much wider issue. In respect of vaginal infections, the study focusses on bacterial vaginosis, trichomonas vaginalis, and candida vaginitis. A range of findings are presented in respect of self-testing. • In respect of common skin conditions, a somewhat different approach appears to have been adopted, relating this to self-diagnosis rather than self-testing. The authors report findings related to the use of questionnaires in the self-diagnosis of eczema, reporting modest sensitivity and specificity, suggesting that the specificity in the range of 0.79-1.00 is “high”. In respect of skin allergy, the focus of self-diagnosis related to self-testing with a self-administered patch-test. • The authors also explore the self-diagnosis of HIV, exploring the self-testing of individuals using a rapid point of care oral fluid test. • Across all these studies, a careful assessment of methodological quality has been included using a standardised approach. • In the discussion, the authors explore the three main areas of investigation, summarising their main findings. There is an exceptionally brief section on study limitations, nothing on study strengths, and nothing about further research that might be appropriate. I have concern that the single sentence summary of the conclusions of the study is not fully justified - it is a sweeping generalisation which seems to confuse the issue of self-diagnosis and self-testing, and misses on what I actually think are actually the most common conditions in primary care. Overall I was therefore, surprised with this review and feel that, whilst there is potentially useful material, it needs to be carefully revisited, refocussed, and substantially reworded prior to being re-presented. The review process itself is fine and well presented – the surrounding material is modest in my view.
--	---

VERSION 1 – AUTHOR RESPONSE

Accuracy of self-diagnosis in conditions commonly managed in primary care: Diagnostic accuracy systematic review and meta-analysis

Reviewer 1	
1. Suggest add a sentence: In the Results, HIV self tests: pls clarify the population group the US HIV study was in, as the pre test probability of HIV varies among different population groups eg Men who have Sex with Men (MSM) compared to general American population.	The following sentence was added: ‘The USA study was conducted in 20 clinical sites, 17 identified as high prevalence sites (2.6%) and 3 as low prevalence sites (0.1%). All studies enrolled participants from the general population including the USA where no breakdown of sexual orientation was reported.’
2. Suggest add a sentence: Equity in testing populations - A sentence or two in the Results would be useful, commenting on whether the self tests	We have added the following sentence: ‘No data were reported by participant

worked the same for everyone or better for some groups. Were there any differences by gender or ethnicity that were noted or striking among the tests? If this information wasn't in the studies, it may be worth mentioning that it was missing.	characteristics such as gender, ethnicity or sexual orientation'
Reviewer 2	
3. I am not clear why you set the object in the study. Really Do self-diagnosis an self-treatment (this is out of scope in the study) is useful strategy for any disease? You should more clearly describe the rationale of the objective with citations.	The reviewer is correct that the debate as to whether it is valid to undertake self-diagnosis of certain conditions is outside the scope of this review. However, it is true this is being considered for a few conditions as evidenced by the studies included in this review. With this in mind, it is important that we assess the accuracy of this type of diagnosis. The background has been extended with additional citations in particular to highlight the controversy over self-diagnosis and the need to assess its efficacy. We have also highlighted that assessing the diagnostic accuracy alone does not inform practice.
4. The review have so comprehensive range, but the findings were limited. You should more clearly limit it by PICO.	We agree that the publication hits from the search were extensive and we expected this as our research question included all common conditions in primary care where self-diagnosis was possible. However, we agree the findings were limited but this was simply because there was a paucity of relevant studies to include. Though this was disappointing, it is a true reflection of the findings from the review given the available evidence this review identified.
5. GP system as UK and Australia is not in global system. if the target is on the GP, you should limit the range of country.	The review is looking at studies in primary care. We have amended the wording in the introduction to reflect this rather than using the term general practice which could be interpreted as UK only. 'The workload in primary care continues to increase' We have added a further reference to confirm this is an issue beyond the UK.

6. L24, The sentence is not clear. Is this a part of methods ? delete ?	We presume the reviewers is referring to the sentence 'We broadly based our review on conditions from this list that are relevant for self-diagnosis' We have deleted the sentence here, and clarified this in the methods (see point 9)
7. L33, How to clarify the study design as "prospective or retrospective" ?	We are unclear how the reviewer wished for us to clarify this further? This study is a systematic review of available evidence.
8. The setting in eligible is not clear. GP only or not ?	The setting is primary care which in the UK includes general practitioner (GP). We have clarified the reference to GP and the PICO to state primary care.
9. The eligible studies have many type of disease, because the search did not clearly set the objects by PICO. You should more clearly set the objects as a review.	We have reworded the 'Type of studies' section to make it clearer by breaking it down into the PICO components. We have added a supplementary table stating the reported 30 commonly managed conditions in primary care.
Reviewer 3	
10. The abstract is structured, concise and well presented. Risk of bias has been assessed in a structured way. Whilst the study sets out to identify "common conditions" in primary care, I was somewhat surprised at the emphasis on self-diagnosis for HIV. There appeared to be no focus on some of the most common conditions encountered in primary care - upper-respiratory infection, urinary tract infection, impetigo etc. The use of comparison with lab reference data may account for some of this, but I had difficulty in respect of "clinical diagnosis" in seeing that the three conditions outlined are indeed worthy of the description attributed. Sophisticated statistics have been applied but the conclusion drawn is surprising, suggesting as it does that 'routine self-diagnosis for common conditions in primary care is not appropriate'.	Thank you for the comments on the abstract and risk of bias. We agree that a number of conditions appear to be missing such as UTIs or upper-respiratory infections. Some studies considered self-testing of these conditions, but did not include self-diagnosis (that is patients interpreting the test results and making a self-diagnosis). We have added an additional sentence to the results section 'No studies of self-diagnosis were found for any other conditions in primary care' In addition, to the discussion section 'Interestingly, no studies of self-diagnosis were

	found for any other conditions in primary care. It was particularly notable that we did not find any studies assessing the diagnostic accuracy of self-diagnosis of common primary conditions such as upper respiratory tract and urinary tract infections.' Our overall conclusion that routine self-diagnosis was not appropriate was predominantly due to the lack of evidence. For two of the conditions few studies meant pooling was inappropriate and so only narrative findings were reported. This provided the reader with an indication of any potential benefit of self-diagnosis, but no more. For HIV, whilst there was sufficient evidence to pool data, the overall finding did not support the use of this test as a rule-out test, and given the clinical consequences of a false negative test result it would not be advisable to promote self-diagnosis without further research. We therefore believe our overall conclusion was not surprising, rather cautionary and based on the evidence presented in the review.
11. Methods – A range of inclusion/exclusion criteria relating to the type of study and focus on clinical diagnosis or laboratory reference standard were included. An appropriate search methodology has been outlined, and would allow for reproducibility of the method. A clear outline of the approach to data collection and analysis, assessment of methodological quality and relevant statistical analysis/data synthesis is provided.	Thank you
12. The results are clearly presented, and well summarised in figure 1. From 5047 records, 18 individual studies were ultimately summarised. Whilst the authors provide the search strategies adopted in each of the databases searched, it is not clear to me exactly how the resulting conditions came to be focussed on.	The conditions described in the review are the only conditions for which studies were identified that reported the diagnostic accuracy of self-diagnosis and therefore met the inclusion criteria of the review. We have added a sentence to the results to clarify this (see point 10)
13. For the conditions investigated, a comprehensive and standardised approach has been adopted to reporting the results of self-diagnosis with relevant laboratory findings. In fact, it looks to me as though in respect of vaginal infections it is self-testing that has been investigated not self-diagnosis which is a much wider issue. In respect of	For vaginal infections, we only included studies where the patient interpreted the results and used the information to self-diagnose. Studies that only reported on self-testing, without self-diagnosis, were excluded and the included studies all reported on self-diagnosis, as the

vaginal infections, the study focusses on bacterial vaginosis, trichomonas vaginalis, and candida vaginitis. A range of findings are presented in respect of self-testing.	objective of the review was the diagnostic accuracy of self-diagnosis. We have restructured the Methods section to clarify this.
14. In respect of common skin conditions, a somewhat different approach appears to have been adopted, relating this to self-diagnosis rather than self-testing. The authors report findings related to the use of questionnaires in the self-diagnosis of eczema, reporting modest sensitivity and specificity, suggesting that the specificity in the range of 0.79-1.00 is “high”. In respect of skin allergy, the focus of self-diagnosis related to self-testing with a self-administered patch-test.	For self-diagnosis of skin allergies, where the patient interpreted the results of the patch themselves this was considered to be self-diagnosis. Without this interpretation, the study was considered to be self-testing and excluded. We hope the restructuring of the methods has clarified this.
15. The authors also explore the self-diagnosis of HIV, exploring the self-testing of individuals using a rapid point of care oral fluid test.	N/A
16. Across all these studies, a careful assessment of methodological quality has been included using a standardised approach	Thank you
17. In the discussion, the authors explore the three main areas of investigation, summarising their main findings. There is an exceptionally brief section on study limitations, nothing on study strengths, and nothing about further research that might be appropriate. I have concern that the single sentence summary of the conclusions of the study is not fully justified - it is a sweeping generalisation which seems to confuse the issue of self-diagnosis and self-testing, and misses on what I actually think are actually the most common conditions in primary care. Overall I was therefore, surprised with this review and feel that, whilst there is potentially useful material, it needs to be carefully revisited, refocussed, and substantially reworded prior to being re-presented. The review process itself is fine and well presented – the surrounding material is modest in my view.	We have added study strengths and further research comments. See point 10 above.

VERSION 2 – REVIEW

REVIEWER	Kerdelmidis, Melissa Canterbury District Health Board, Planning & Funding
--

REVIEW RETURNED	07-Jul-2022
-------------

GENERAL COMMENTS	1.It is now clearer why these 3 particular conditions and their tests were selected. While HIV is not the most common condition in most general practices, it was one of the few conditions for which studies were available which met your inclusion criteria. 2.Abstract: Please add a sentence to the abstract to clarify there was only suitable information about 3 conditions, so that's why you chose the 3 to focus on. [It is clearer later in the paper now]. 3.To be more accurate, the abstract conclusion should also mention that self-diagnosis was not suitable for the 3 conditions in primary care which you reviewed. 4.Discussion: Suggest mention that if this protocol was repeated, it is likely rapid antigen covid tests would feature, as they have become quite commonly used at home over 2020-2.
---

REVIEWER	Campbell, John University of Exeter, Primary Care
-----------------	--

REVIEW RETURNED	10-Aug-2022
-------------

GENERAL COMMENTS	Thank you for the opportunity to revisit this paper including the response to reviewers comments and tracked change versions; the authors have taken on board the earlier observations in a careful, systematic, and conscientious way and made many useful clarifying amendments and added additional text to the manuscript. A range of relevant and interesting findings are presented and I have no hesitation in recommending publication.
---

VERSION 2 – AUTHOR RESPONSE

Reviewer 1	
18. It is now clearer why these 3 particular conditions and their tests were selected. While HIV is not the most common condition in most general practices, it was one of the few conditions for which studies were available which met your inclusion criteria.	Thank you
19. Abstract: Please add a sentence to the abstract to clarify there was only suitable information about 3 conditions, so that's why you chose the 3 to focus on. [It is clearer later in the paper now].	We updated the Methods & Results section in the abstract to read: Searches identified 5,047 records resulting in 18 included studies covering the self-diagnosis of three common conditions: vaginal infection (five studies), common skin conditions (four studies) and HIV (nine studies). No studies were found for any other condition.
20. To be more accurate, the abstract conclusion should also mention that self-diagnosis was not suitable for the 3	We have updated the conclusion to read:

conditions in primary care which you reviewed.	The current evidence does not support routine self-diagnosis for vaginal infections, common skin conditions and HIV in primary care.
21. Discussion: Suggest mention that if this protocol was repeated, it is likely rapid antigen Covid tests would feature, as they have become quite commonly used at home over 2020-22	Thank you for this suggestion. We have added additional text in the discussion: As technology develops potentially enabling increased self-diagnosis in primary care, we would expect future reviews examining this research question to include more common conditions. In particular, we would expect to include studies for self-diagnosis of COVID-19 following the rapid development of tests during the COVID-19 pandemic.
Reviewer 2	
Thank you for the opportunity to revisit this paper including the response to reviewers comments and tracked change versions; the authors have taken on board the earlier observations in a careful, systematic, and conscientious way and made many useful clarifying amendments and added additional text to the manuscript. A range of relevant and interesting findings are presented and I have no hesitation in recommending publication.	Thank you
Comment from the Editor	
22. Please explain in the methods section how the risk of bias chart is meant to be interpreted.	We have updated the Assessment of methodological quality in the methods section to read: We used the Quality Assessment of Diagnostic Accuracy Studies-2 (QUADAS-2) tool to assess methodological quality of included studies. This considered the risk of bias in four domains (patient selection, index test, reference standard, flow and timing), as well as assessing the applicability (for the first three domains) of the studies to the review research question. Studies were assessed as low, high or unclear risk of bias/concerns regarding applicability for each domain. Two reviewers (JM, AP) independently assessed studies' methodological quality; disagreements were resolved by discussion, or if necessary, by a third reviewer. The results of the QUADAS-2 assessment were presented in a summary table.